# The Impact of Repeated Attachment Priming on Paranoia, Mood and Help-Seeking Intentions in an Analogue Sample

**DOI:** 10.3390/brainsci11101257

**Published:** 2021-09-22

**Authors:** Katherine Newman-Taylor, Monica Sood, Angela C. Rowe, Katherine B. Carnelley

**Affiliations:** 1Psychology Department, University of Southampton, Shackleton Building, Highfield Campus, Southampton SO17 1BJ, UK; ms13g14@soton.ac.uk (M.S.); Kc6@soton.ac.uk (K.B.C.); 2School of Psychological Science, University of Bristol, 12A Priory Road, Bristol BS8 4PT, UK; a.c.rowe@bristol.ac.uk

**Keywords:** paranoia, mood, affect, help seeking, cognitive fusion, attachment, avoidant attachment, security priming

## Abstract

Attachment security priming effects therapeutic change in people with depression and anxiety. Preliminary studies indicate that visualising secure attachment memories also reduces paranoia in non-clinical and clinical groups, probably due to a decrease in cognitive fusion. Benefits to clinical populations depend on the sustainability of these effects and the impact on help-seeking behaviours. The combination of paranoia and an insecure-avoidant attachment style is likely to be a particular barrier to help seeking. We used a longitudinal experimental design to test the impact of repeated attachment priming on paranoia, mood and help-seeking intentions and whether cognitive fusion mediates these effects. Seventy-nine people with high levels of non-clinical paranoia, aged 18–50 years (M = 20.53, SD = 4.57), were randomly assigned to a secure or insecure-avoidant priming condition. Participants rehearsed the visualisation prime on four consecutive days and were assessed on standardised measures of paranoia, positive and negative affect, help-seeking intentions and cognitive fusion. A series of mixed-model analyses of variance showed that security priming decreases paranoia, negative affect and cognitive fusion and increases positive affect and help seeking, compared to insecure-avoidant priming. Examining the impact of primed attachment (rather than measured attachment style) allows us to draw conclusions about the causal processes involved; mediation analyses showed indirect effects of the primes on paranoia and negative affect through cognitive fusion. With a growing understanding of (1) the impact of security priming on paranoia, affect and help-seeking behaviours, (2) causal mechanisms and (3) sustainability of effects, security priming may be developed into a viable intervention for clinical populations.

## 1. Introduction

### 1.1. Attachment Theory

Attachment theory provides a useful framework to conceptualise mental health. Bowlby [1] proposed that humans are innately predisposed to attach to others and that long-term attachment relationships form the basis of ‘internal working models.’ Consistent experiences (e.g., of care or rejection) are abstracted and internalised to form chronically activated (and chronically applied) behavioural templates that guide emotion regulation and relationship-relevant cognition and behaviour. Individual differences in internal working models are referred to as ‘attachment styles’ which are important predictors of close relationship behaviour, relationship quality and mental health outcomes [2]. Attachment styles are activated in relationship-relevant or threatening situations and drive style-congruent thinking, feeling and behaviour [1,3]. These trait-like patterns are commonly conceptualised along two dimensions: anxiety regarding abandonment and avoidance of intimacy [4]. Individuals can be low or high on each dimension. Individuals high on either attachment dimension are described as being attachment insecure. Those low on both dimensions are described as being attachment secure.

Individuals who are high in attachment avoidance attempt to quash positive and negative emotions. They are likely to ignore or deny threats to the self (including to attachment relationships) and show compulsive self-reliance in most aspects of their lives. They assume that they do not need others, who are perceived as unreliable and untrustworthy. Individuals high in attachment anxiety (or ‘preoccupation’) find it hard to regulate their emotions and fear abandonment. They tend to see the self as unworthy of love and care from others and see others as unreliable. By contrast, secure individuals show optimal emotion regulation. They find it easy to trust and depend on others, see the self as worthy of care and love and view others as available when needed. Attachment security is a protective factor against mental ill-health and has a number of well-established wellbeing correlates, while attachment insecurity acts as a vulnerability factor for a wide range of psychopathologies including depression, anxiety [5] and paranoia—unfounded interpersonal threat beliefs characteristic of diagnoses such as schizophrenia [6].

In addition to a dispositional (or trait) attachment style, adults hold relationship-specific patterns based on previous attachment relationships; even an individual who is generally high in anxiety or avoidance is likely to have had attachment relationships characterised by security, and so develop a secure internal working model [3,7,8]. It is helpful to think of attachment styles organised hierarchically, with a person’s dispositional style, which is most accessible (due to frequent use and reinforced neural pathways), at the top of the hierarchy, and relationship-specific attachment styles (e.g., to either parent, romantic partner, particular siblings) less readily available and therefore lower on the hierarchy [9].

### 1.2. Attachment Security Priming

Relationship-specific attachment styles can be activated, or ‘primed’, which has the effect of rehearsing this neural pathway. Once activated, primed attachment styles drive attachment-relevant information processing, feelings and behaviour in the same way as the dispositional attachment style. Importantly, priming attachment security produces security-congruent thoughts, feelings and behaviour irrespective of dispositional style [3,8,10].

Security priming techniques commonly involve asking participants to think and write about a previous or current relationship characterised by a particular attachment style. By activating this mental representation, the attachment style pertaining to that relationship is also rendered active [3]. Security priming techniques are robust, reliable and distinct from positive mood manipulations [11].

Security priming has been shown to enhance a range of desirable outcomes, including positive mood, self-views and relationship expectations, empathy, generosity and caregiving [8,10,12]. Importantly from a clinical perspective, security priming can reduce anxiety and depression in clinical and non-clinical populations [13,14,15,16]. Furthermore, if repeatedly delivered, priming effects can be sustained for days in non-clinical and clinical samples [12,17,18]. In a systematic review of this large body of literature, Rowe et al. [19] showed that attachment security priming effectively attenuates negative affect and increases positive affect.

### 1.3. Attachment Style Priming in Paranoia

Paranoia describes a continuum of unfounded interpersonal threat beliefs, ranging from transient suspicious thoughts (common in the general population) to enduring persecutory delusions whereby the person is convinced that others are intending them harm (more typical of clinical groups) [20,21]. The continuum model assumes that severe forms of paranoia develop from the mechanisms that maintain non-clinical paranoia and that isolating and targeting these in analogue groups (with elevated levels of non-clinical paranoia) can inform the development of psychological interventions for clinical populations [21].

Despite now well-established associations between paranoia and both attachment anxiety and avoidance in non-clinical [22,23,24], ‘ultra high risk’ (help-seeking individuals with attenuated or brief, limited psychotic symptoms who do not meet criteria for psychosis diagnoses [25]) and clinical groups [24,26,27,28,29], limited research has focused on the corollary—that fostering a sense of ‘felt security’ could attenuate interpersonal fears (i.e., paranoia).

Initial studies investigating the benefits of attachment security priming for paranoia in analogue groups [30,31,32,33] and preliminary clinical samples [34,35] show that security priming reduces state paranoia and negative affect and increases positive affect and self-esteem. Importantly, security priming reduces state paranoia for people high in trait attachment avoidance and for people high in trait attachment anxiety [32]. We do not yet know if these effects can be sustained over time and have yet to confirm the cognitive mechanisms by which security priming impacts paranoia.

Cognitive fusion [36] is a candidate causal mechanism which may account for the effects of security priming in people with elevated levels of paranoia. Cognitive fusion describes the extent to which we are able to ‘decentre’ [37] or distance ourselves [38] from internal experience. The inability to step back from thoughts, feelings and urges and recognise these as transient events of the mind (rather than necessarily accurate reflections of reality) is associated with psychopathology transdiagnostically [39,40,41,42]. Initial studies examining attachment priming in people with high levels of non-clinical paranoia suggest that cognitive fusion and self- and other-beliefs mediate the impact on state paranoia and affect with large effects for fusion and medium effects for beliefs [32,33]. Cognitive fusion appears to be a key mechanism by which security priming reduces paranoia and negative affect.

Together, these studies show that attachment security priming reduces paranoia and distress [30,31,32,33,34,35] and that cognitive fusion is a mediating mechanism in the attachment–paranoia association [32,33].

### 1.4. Impact of Attachment Style on Help Seeking in Paranoia

Priming research in paranoia has focused on cognitive and affective outcomes to date. Help seeking is a key behavioural outcome in psychosis research, given the damaging effects of ‘duration of untreated psychosis’ (DUP). Psychosis is an umbrella term for mental health conditions typically associated with diagnoses such as schizophrenia and schizoaffective disorder, involving unusual perceptual experiences (e.g., hearing voices), firmly held unshared beliefs (delusions), difficulty thinking and concentrating (‘thought disorder’) and becoming withdrawn and inexpressive (‘negative symptoms’) [43].

Typically, people wait one to two years following initial signs of psychosis before accessing services [44,45], leading to significant treatment delays and poorer prognosis [46,47]. DUP predicts severity of illness and likelihood of recovery several years later, resulting in considerable human, societal and healthcare costs [48,49]. For these reasons, the World Health Organisation identifies DUP as an international healthcare priority [50,51]. Help-seeking delays account for a third of DUP [52]. Understanding the reasons why individuals delay help seeking (alongside the causes of service delays) is likely to inform efforts to reduce DUP in clinical populations [53].

Attachment style predicts help seeking in clinical populations with severe mental health problems including psychosis. Attachment avoidance is associated with fewer help-seeking attempts, less self-disclosure and increased likelihood of rejecting help offered, whereas attachment anxiety/preoccupation is associated with increased help seeking and self-disclosure [54]. Attachment-avoidant behaviours are therefore likely to delay initial contact with services. In people with psychosis, attachment insecurity is associated with interpersonal problems and difficulties in therapeutic relationships [55] which may disrupt early contacts, delaying treatment further. While these studies do not examine outcomes by symptom clusters (e.g., paranoia), it is highly likely that interpersonal threat beliefs interfere with help seeking [56]. The combination of paranoia and an avoidant attachment style, characterised by compulsive self-reliance and the assumption that others are unreliable and untrustworthy, may be a particular barrier to help seeking. We therefore need to examine the causal effects of priming attachment security and avoidance on help seeking for people with paranoia.

To date, just one study has assessed the impact of attachment style priming on behaviour; Sood et al. [32] found that secure priming increased help-seeking intentions (though this was not mediated by cognitive fusion or negative beliefs about self or others). This is a promising but unreplicated study.

### 1.5. Current Study

In summary, we know that secure attachment priming is effective in reducing paranoia and distress and that cognitive fusion is likely to be a key mediating factor. We do not yet know if these effects can be sustained, and just one study [32] has examined the impact on help seeking. We used a longitudinal experimental design to test the impact of repeated attachment security priming on paranoia, mood and help-seeking intentions, and whether cognitive fusion mediates these effects. The use of repeated primes in a high non-clinical paranoia sample is novel, and examination of the impact on help-seeking intentions requires replication. Additionally, prior research in this area has not focused on positive and negative mood as outcomes or the role of cognitive fusion as an explanatory variable in these relationships. We compared the impact of secure vs. insecure-avoidant primes, given likely differences in help seeking associated with these styles [54,55].

We hypothesised that secure attachment priming would result in lower levels of paranoia and negative affect and higher levels of positive affect and help-seeking intentions, compared with insecure-avoidant priming, and that these effects would be sustained over time:Relative to insecure-avoidant attachment priming, secure attachment priming will reduce state paranoia, negative affect and cognitive fusion and increase state positive affect and help-seeking intentions from pre-prime (Time 1) to post-prime (Times 1b, 3, 5).Relative to the insecure-avoidant group, the secure attachment group will have lower levels of state paranoia, negative affect and cognitive fusion and higher levels of state positive affect and help-seeking intentions, post attachment priming (at Times 1b, 3, 5).These changes will be mediated by cognitive fusion – cognitive fusion will mediate the impact of attachment priming (insecure-avoidant vs. secure) on (a) paranoia, (b) negative affect, (c) positive affect and (d) help seeking; relative to insecure-avoidant priming, secure priming will (a) decrease paranoia, (b) decrease negative affect, (c) increase positive affect and (d) increase help-seeking intentions, via decreased cognitive fusion.

## 2. Materials and Methods

### 2.1. Ethical Considerations

This study received ethical approval from the University of Southampton, UK (ID: 30332), and was pre-registered on the Open Science Framework (https://osf.io/5yebw).

### 2.2. Design

We used a longitudinal mixed-model experimental design with one between-participants variable (prime: insecure-avoidant vs. secure) and one within-participants variable (time: pre-prime (Time 1a) vs. post-prime (Times 1b, 3, and 5))—see Figure 1. The dependent variables were state paranoia, mood, help-seeking intentions and cognitive fusion. Trait paranoia, mood, cognitive fusion and attachment orientation were measured prior to the experimental manipulation to check whether the groups were comparable on these key variables. 

### 2.3. Participants

We recruited adults with high levels of non-clinical paranoia from open research, social media and university research websites. Participants were screened using the Paranoia Scale [57] and invited to participate if they scored at or above 53 (one SD above the mean of the standardisation sample), following Bullock et al. [30]. Seventy-nine participants completed the study, aged 18 to 50 years (M = 20.53, SD = 4.57). Sixty-nine (87.3%) were students. The majority were female (65; 82.5%) and identified as British (43; 54.4%); others self-reported as other white background (11; 13.9%), Indian (6; 7.6%), African (4; 5.1%), Bangladeshi (2; 2.5%), Caribbean (1; 1.3%), Pakistani (2; 2.5%), White and Asian (1; 1.3%), White and Black Caribbean (1; 1.3%), other Asian background (2; 2.5%) and other mixed/multiple ethnic background (4; 5.1%).

### 2.4. Procedure

The study was advertised and made available on open research websites accessed by the general population (Psychological Research on the Net), social media platforms (Reddit, Facebook, LinkedIn and Twitter) and the University of Southampton research website open to students. The entire study was conducted online. Following informed consent, participants completed the Paranoia Screen [57] to determine eligibility. One week later, those reporting high levels of non-clinical paranoia were asked to provide demographic information and complete other trait measures (mood, cognitive fusion and attachment) and all state measures (paranoia, mood, help-seeking and cognitive fusion). They were then randomly allocated (using an online randomiser) to the secure or insecure-avoidant condition in which they were asked to visualise and then write about either a secure or insecure-avoidant attachment relationship for ten minutes (following Bartz & Lydon [58]). Participants repeated the state measures and manipulation checks while instructed to hold the visualisation in mind. On each of the following four days, participants received an email prompting them to rehearse the visualisation and complete the writing task for three minutes. The state measures were repeated after the prime on the third day. Participants repeated the state and trait measures and a mood repair following the prime on the fifth day. Finally, participants were debriefed and thanked.

### 2.5. Measures

#### 2.5.1. Trait Measures

Paranoia: The Paranoia Scale (PS [57]) is a 20-item measure designed to assess sub-clinical trait paranoia. Participants rated statements on a 5-point scale (1 = ‘not at all applicable to me’ to 5 = ‘extremely applicable to me’). The PS has acceptable internal reliability (α = 0.72), which was excellent in the screened sample (N = 664; α = 0.92).

Mood: The Depression, Anxiety and Stress Scale (DASS [59]) is a 42-item measure yielding three subscales—depression, anxiety and stress. Participants rated applicability of items over the past week on a 4-point scale (0 = ‘did not apply to me at all’ to 3 = ‘applied to me very much, or most of the time’). The subscales have good to excellent internal consistency (αs > 0.84), and these were excellent for the current sample at both time points (αs ≥ 0.90).

Cognitive fusion: The short Cognitive Fusion Questionnaire (CFQ-7 [38]) is a 7-item measure of the extent to which people tend to be fused with their thoughts. Participants rated items on a 7-point scale (1 = ‘never true’ to 7 = ‘always true’). Internal consistency is excellent (α = 0.90), and this was excellent for the current sample at both time points (αs < 0.90).

Attachment orientation: The Experiences in Close Relationships questionnaire (ECR [4] adapted by Rowe & Carnelley [8]) is a 36-item measure of anxious and avoidant attachment dimensions. The measure was adapted to elicit responses regarding close others generally. The attachment anxiety and avoidance subscales (each of 18 items) were rated on a 7-point scale (1 = ‘disagree strongly’ to 7 = ‘agree strongly’). The measure has excellent internal consistency for both subscales (αs > 0.90) [5], and these were good to excellent in the current sample at both time points (αs > 0.88).

#### 2.5.2. State Measures

Paranoia: The Adapted Paranoia Checklist (APC [60]) is a 5-item scale of current paranoid ideation. Items were rated ‘at the moment’ on a 11-point scale (0 = ‘not at all’ to 10 = ‘very strongly’). The 5-item APC has excellent internal consistency (α = 0.90), and this was adequate in the current sample across all time points (αs > 0.74).

Mood: The Positive and Negative Affect Schedule (PANAS [61]) yields positive and negative affect subscales (each of 10 items). Items were rated on a 5-point scale (1 = ‘very slightly/not at all’ to 5 = ‘extremely’). The measure has good to excellent internal consistency (positive: α = 0.86–0.90; negative: α = 0.84–0.87) which was good to excellent in the current sample across all time points (positive: αs > 0.92; negative: αs > 0.86).

Help seeking: The Help-Seeking Measure-State (HSM-S [32]) is a 3-item questionnaire assessing help-seeking intentions. Participants rated the likelihood of seeking help if feeling upset ‘right now’ on a 5-point scale (1 = ‘not at all’ to 5 = ‘extremely’). Internal consistency in the original sample was good to excellent (α = 0.89–0.93), and this was good to excellent in the current sample across all time points (αs > 0.89).

Cognitive fusion: The State Cognitive Fusion Questionnaire (CFQ-S [62]) is a 7-item measure of fusion with thoughts ‘at this moment.’ Participants rated items on a 7-point scale (1 = ‘completely untrue’ to 7 = ‘completely true’). Internal consistency is excellent (α = 0.90), and this was also excellent for the current sample across all time points (αs > 0.92).

#### 2.5.3. Experimental Manipulation

Priming instructions: We used the Bartz and Lydon [58] attachment priming procedure. Participants are asked to bring to mind a close and comfortable relationship (secure) or one in which they feel uncomfortable being too close to the other person and struggle to trust them (insecure-avoidant). Participants visualised the relationship while writing about this and linked feelings for ten minutes. Participants were instructed to hold the visualisation in mind while repeating the manipulation checks and repeated state measures.

Manipulation checks: Participants rated the vividness of the image evoked on a 10-point scale (1 = ‘not at all vivid’ to 10 = ‘extremely vivid’) and the percentage of time the image was held in mind while completing the measures (0–100%). They also completed the 10-item Felt Security measure [63] of the extent to which thinking about the person in the image elicited elements of secure attachment (e.g., comforted, safe, secure) on a 6-point scale (1 = ‘not at all’ to 6 = ‘very much’). Internal consistency for the Felt Security measure is excellent (α > 0.90) [32,33], and this was also excellent for the current sample (α = 0.98).

#### 2.5.4. Data Analyses

We analysed the data using IBM SPSS 27 for Windows. Following Tabachnick and Fidell [64], we excluded participants with more than 5% missing data (*n* = 4) and replaced all other missing data (<5%) with the participant mean (*n* = 1). Data were normal (not severely skewed or kurtotic), and no univariate outliers (z > ±3.29) were observed for any variable except for baseline trait cognitive fusion.

We first tested for pre-manipulation differences between the two groups, using one-way analyses of variance (ANOVAs) to examine age and trait measures and chi-squared for gender. We then used ANOVAs to test predicted differences in the two groups following experimental manipulation. Independent Variables (IVs): attachment prime (2 levels—secure vs. insecure-avoidant) and time (4 levels—Time 1a, 1b, 3 and 5). State Dependent Variables (DVs): paranoia (APC), positive and negative affect (PANAS), help seeking (HSM-S) and cognitive fusion (CFQ-state). Sood et al. [32] found an effect size of 0.17 for the prime condition by time interaction; a G*Power analysis indicated that to detect an effect size of 0.17 at *p* = 0.05, with 95% power, with two groups at four time points, 76 participants would be required (we had 79 participants).

We used PROCESS version 3 [65] to examine whether cognitive fusion mediated the association between attachment priming and paranoia, positive and negative affect and help-seeking. IV: attachment prime (2 levels—insecure-avoidant vs. secure, dummy coded). DVs: difference scores for state paranoia, positive and negative affect and help seeking (Time 5—Time 1a). Mediator: difference score for cognitive fusion (Time 5—Time 1a). We infer indirect effects using percentile bootstrapping with 5000 bootstrapped samples (Preacher & Hayes, 2004)—this produces a 95% confidence interval (CI) for each indirect effect. When the CI does not straddle zero, mediation is observed. Following Hayes (2018), we report partially standardized indirect effects (abps): small (0.02), medium (0.15), large (0.40) [66].

## 3. Results

### 3.1. Demographic and Trait Measures

Demographic and trait information are shown in Table 1. We compared the two conditions for initial differences using *t*-tests and a chi-squared analysis for gender. There were no initial differences in age (F(1, 76) = 2.81, *p* = 0.10), gender (X2(2, N = 79) = 0.07, *p* = 0.79), trait paranoia (F(1, 76) = 1.90, *p* = 0.17), mood (F(1, 76) = 0.37, *p* = 0.55), cognitive fusion (F(1, 76) = 2.82, *p* = 0.10), attachment anxiety (F(1, 76) = 1.82, *p* = 0.18) or attachment avoidance (F(1, 76) = 0.34, *p* = 0.56). See Appendix A for intercorrelations between trait variables (Appendix A).

### 3.2. Manipulation Checks

There were no differences in vividness of the prime (F(1, 76) = 0.21, *p* = 0.65) or time holding the prime in mind (F(1, 71) = 0.88, *p* = 0.35). Differences in felt security (F(1, 77) = 0.47, *p* = 0.14, n2 = 0.38) indicate that the experimental manipulation was successful; participants in the secure condition scored higher (M = 42.67, SD = 14.62) than those in the insecure-avoidant condition (M = 22.08, SD = 11.61) following the prime.

### 3.3. Impact of Attachment Priming

Table 2 shows state measures before and after prime manipulation for the two groups. See Appendix A for intercorrelations between state variables (Appendix A).

### 3.4. State Paranoia

There was no main effect of prime (F(1, 77) = 0.00, *p* = 0.99), though there was a main effect of time (F(1, 77) = 20.00, *p* < 0.001, np2 = 0.21) and a prime by time interaction (F(1,77) = 19.30, *p* < 0.001, np2 = 0.20) (see Figure 2). Simple effects showed that the two groups did not differ at time 1a (F(1, 77) = 3.71, *p* = 0.06), time 1b, (F(1, 77) = 0.19, *p* = 0.67), time 3 (F(1, 77) = 1.05, *p* = 0.31) and time 5 (F(1, 77) = 1.25, *p* = 0.27). Post hoc paired t-tests showed that in the secure prime condition, paranoia decreased from time 1a to 1b (*t*(41) = 3.43, *p* = 0.001, d = 0.24), time 1b to 3 (*t*(41) = 2.97, *p* = 0.005, d = 0.26) and time 3 to 5 (*t*(41) = 2.29, *p* = 0.03, d = 0.18). In the insecure-avoidant prime condition, paranoia did not change from time 1a to 1b (*t*(36) = −0.48, *p* = 0.64), time 1b to 3 (*t*(36) = −0.61, *p* = 0.55) and time 3 to 5 (*t*(36) = 1.44, *p* = 0.16).

### 3.5. State Mood

Positive affect: There was no main effect of prime (F(1, 77) = 0.20, *p* = 0.66) or time (F(1, 77) = 1.30, *p* = 0.26), though there was a prime by time interaction (F(1, 77) = 11.89, *p* = 0.001, np2 = 0.13) (see Figure 2). The two groups did not differ at time 1a (F(1, 77) = 1.76, *p* = 0.19), time 1b, (F(1, 77) = 0.04, *p* = 0.84), time 3 (F(1, 77) = 1.82, *p* = 0.18) and time 5 (F(1, 77) = 1.66, *p* = 0.20). In the secure prime condition, positive affect did not change from time 1a to 1b (*t*(41) = 0.35, *p* = 0.73), time 1b to 3 (*t*(41) = −1.45, *p* = 0.15) and time 3 to 5 (*t*(41) = −0.60, *p* = 0.55). In the insecure-avoidant prime condition, positive affect decreased from time 1a to 1b (*t*(36) = 3.97, *p* < 0.001, d = 0.37) but did not change from time 1b to 3 (*t*(36) = 0.71, *p* = 0.49) and time 3 to 5 (*t*(36) = −0.28, *p* = 0.78).

Negative affect: There was no main effect of prime (F(1, 77) = 0.93, *p* = 0.34), though there was a main effect of time (F(1, 77) = 3.90, *p* = 0.01, np2 = 0.05) and a prime by time interaction (F(1, 77) = 3.21, *p* = 0.02, np2 = 0.04) (see Figure 2). The two groups did not differ at time 1a (F(1, 77) = 0.05, *p* = 0.83), time 3 (F(1, 77) = 0.38, *p* = 0.54) or time 5 (F(1, 77) = 0.83, *p* = 0.37), though differed at time 1b (F(1, 77) = 4.03, *p* = 0.05, n2 = 0.05); compared with the avoidant group, the secure prime group reported lower levels of negative affect at time 1b. In the secure prime condition, negative affect decreased from time 1a to 1b (*t*(41) = 3.99, *p* = 0.001, d = 0.34) but did not change from time 1b to 3 (*t*(41) = −1.70, *p* = 0.10) or time 3 to 5 (*t*(41) = 1.84, *p* = 0.07). In the insecure-avoidant prime condition, negative affect increased from time 1a to 1b (*t*(36) = −2.03, *p* = 0.05, d = 0.18) but did not change from time 1b to 3 (*t*(36) = 1.05, *p* = 0.30) or time 3 to 5 (*t*(36) = 1.89, *p* = 0.07).

### 3.6. State Help-Seeking

There was no main effect of prime (F(1, 77) = 0.03, *p* = 0.86), a main effect of time (F(1, 77) = 6.43, *p* < 0.001, np2 = 0.08), and a prime by time interaction (F(1, 77) = 5.29, *p* = 0.002, np2 = 0.06) (see Figure 2). The two groups did not differ at time 1a (F(1, 77) = 1.22, *p* = 0.27), time 1b, (F(1, 77) = 0.05, *p* = 0.82), time 3 (F(1, 77) = 2.17, *p* = 0.15) and time 5 (F(1, 77) = 0.003, *p* = 0.96). In the secure prime condition, help seeking increased from time 1a to 1b (*t*(41) = −2.77, *p* = 0.008, d = 0.23) and time 1b to 3 (*t*(41) = −3.11, *p* = 0.003, d = 0.22) but did not change from time 3 to 5 (*t*(41) = 0.73, *p* = 0.47). In the insecure-avoidant prime condition, help seeking did not change from time 1a to 1b (*t*(36) = 1.00, *p* = 0.32) and time 1b to 3 (*t*(36) = 0.31, *p* = 0.76) and (unexpectedly) increased from time 3 to 5 (*t*(36) = −2.04, *p* = 0.05, d = 0.25).

### 3.7. State Cognitive Fusion

There was no main effect of prime (F(1, 77) = 0.41, *p* = 0.53), though there was a main effect of time (F(1, 77) = 5.25, *p* = 0.002, np2 = 0.06) and a prime by time interaction (F(1, 77) = 3.57, *p* = 0.02, np2 = 0.04) (see Figure 2). The two groups did not differ at time 1a (F(1, 77) = 1.06, *p* = 0.31), time 1b, (F(1, 77) = 2.02, *p* = 0.16), time 3 (F(1, 77) = 0.53, *p* = 0.47) and time 5 (F(1, 77) = 1.09, *p* = 0.30). In the secure prime condition, cognitive fusion decreased from time 1a to 1b (*t*(41) = 3.66, *p* = 0.001, d = 0.33), did not change from time 1b to 3 (*t*(41) = −0.02, *p* = 0.16, d = 0.99) and trended toward decreasing from time 3 to 5 (*t*(41) = 1.93, *p* = 0.06). In the insecure-avoidant prime condition, cognitive fusion increased from time 1a to 1b (*t*(36) = −2.27, *p* = 0.03, d = 0.22) but did not change from time 1b to 3 (*t*(36) = 1.44, *p* = 0.16) or time 3 to 5 (*t*(36) = 1.48, *p* = 0.15).

### 3.8. Mediation

We performed four mediation analyses to test whether cognitive fusion mediates the effect of attachment prime (insecure-avoidant vs. secure) on state paranoia, positive and negative affect and help seeking.

Paranoia: There was a direct effect of attachment prime (insecure-avoidant vs. secure) on paranoia (see Figure 3). Independent of prime condition, greater levels of paranoia were observed in participants who reported higher levels of cognitive fusion. There was also an indirect effect of prime condition on paranoia through cognitive fusion (ab = −1.46, SE = 0.74, 95% CI = (−3.06, −2.12)) with a medium effect (abps = −0.21, SE = 0.10, 95% CI = (−0.42, −0.03)). Relative to the insecure-avoidant prime, the secure prime decreased fusion which, in turn, decreased paranoia.

Positive affect: There was a direct effect of attachment prime on positive affect, though cognitive fusion did not predict positive affect, and there was no indirect effect of prime condition on paranoia through cognitive fusion (ab = 0.72, SE = 0.78, 95% CI = (−0.45, 2.68)).

Negative affect: There was no direct effect of attachment prime on negative affect (see Figure 3). Independent of prime condition, higher levels of negative affect were observed in participants who reported higher levels of cognitive fusion. There was an indirect effect of prime condition on negative affect through cognitive fusion (ab = −2.11, SE = 0.99, 95% CI = (−4.16, −0.29)) with a medium effect (abps = 0.28, SE = 0.12, 95% CI = (−0.53, −0.04)). Relative to the insecure-avoidant prime, the secure prime decreased fusion which, in turn, decreased negative affect.

Help seeking: There was no direct effect of attachment prime on help seeking, and cognitive fusion did not predict help-seeking. There was no indirect effect of prime condition on help seeking through cognitive fusion (ab = 0.10, SE = 0.10, 95% CI = (−0.03, 0.35)).

See Appendix A for correlations between state difference score variables in mediation (3).

### 3.9. Supplementary Analyses

As the majority of participants were women aged 18–26, we reanalysed the data with this sub-sample. We found the same pattern of results, though some lost significance at *p* < 0.05%, as follows: (a) we found trends for the main effect of time for negative affect (*p* = 0.07) and the between-group difference in negative affect at time 1b (*p* = 0.07); (b) the imagery groups differed on help seeking at time 3 (F(1, 60) = 4.93, *p* = 0.03, np2 = 0.08), with the secure-primed group reporting more help-seeking intentions than the avoidant-primed group; (c) in the avoidant-primed condition, there was a trend for the change in help-seeking from time 3 to 5 (*t*(28) = −1.75, *p* = 0.09); (d) there was a trend for the condition by time interaction on cognitive fusion (F(1, 60) = 2.17, *p* = 0.09) and a change in cognitive fusion from time 1a to 1b in the avoidant-primed condition (*t*(28) = −1.74, *p* = 0.09); and (e) there were no indirect effects of imagery condition on paranoia (ab = −0.90, SE = 0.82, 95% CI = (−2.70, 0.66)) and negative affect (ab = −1.13, SE = 0.97, 95% CI = (−3.09, 0.83)) via cognitive fusion. Different results from those with the full sample are likely to be due to low statistical power for the sub-sample. We also conducted the analyses excluding just the one participant aged 50; the pattern of results was the same as the original results.

Additionally, see Appendix A for exploratory analyses of the impact of repeated priming on trait variables (4) and change in trait attachment anxiety from baseline (time 1a) to post-prime (time 5) (5).

## 4. Discussion

In this study, we aimed to determine whether repeated attachment priming affects state paranoia, mood and help-seeking intentions in people with high trait paranoia, and if cognitive fusion mediates these effects. As predicted, secure attachment priming resulted in lower levels of paranoia and negative affect and higher levels of positive affect and help-seeking intentions, compared with insecure-avoidant priming. The hypotheses were partially supported, though not all variables differed by group at each time point. The overall pattern of results suggests that the secure prime resulted in predicted effects over time, whereas the impact of the avoidant prime was lost after day 1 (which may account for the unexpected increase in help seeking in this group from time 3 to 5).

The manipulation checks showed that both conditions elicited comparably vivid images, which were held in mind for the same amount of time and were successful in manipulating felt security (which was greater in the secure than the insecure-avoidant condition). Cognitive fusion mediated the priming effects (insecure-avoidant vs. secure) on paranoia and negative affect but not on positive affect or help-seeking. Relative to the insecure-avoidant prime, the secure prime decreased cognitive fusion which, in turn, decreased paranoia and negative affect.

These results add to the growing body of literature demonstrating that security priming reduces paranoia and negative affect in people with elevated and clinical levels of paranoia and the role of potential causal mechanisms [31,32,33,34,35]. This is the first study to demonstrate the effects of repeated priming in a sample with high trait paranoia and replicates initial evidence of the impact on help-seeking intentions in this group [32].

The reduction in state paranoia at each time point shows that repeated security priming can have a cumulative effect which is sustained and strengthened with each practice. Replication of the finding that security priming affects help seeking as well cognitive and affective outcomes (originally shown by Sood et al. [32]) is important, given the likely impact of both attachment avoidance and paranoia on help-seeking, which in turn lengthens DUP with serious personal, societal and healthcare costs [48,49,50,51].

Our study design allows us to draw conclusions about the causal impact of attachment on paranoia and candidate processes involved in this relationship. By examining primed attachment (rather than measured attachment style), we can be confident that the secure prime decreased cognitive fusion, which caused the reduction in paranoia and negative affect. This is the first study to show that cognitive fusion is a strong predictor of negative affect (as opposed to anxiety specifically) in people with non-clinical paranoia. This is consistent with the suggestion that the inability to ‘step back’ from internal experience is associated with psychopathology transdiagnostically [39]. Facilitating ‘defusion’ from compelling internal thoughts and beliefs may therefore be a helpful method of reducing distress in people with high levels of paranoia.

The finding that cognitive fusion did not mediate the impact of primes on positive affect is novel and requires replication. Although high cognitive fusion predicts decreased negative affect, low cognitive fusion does not necessarily predict increased positive affect. Interestingly, there is some evidence that emotion regulation interventions in clinical samples reduce negative affect but do not increase positive affect [67,68]. The finding that cognitive fusion did not mediate the impact on help seeking is consistent with Sood et al. [32] who highlight the possible role of other factors (such as availability of help-offering others). This now requires investigation.

Despite seeking to recruit from the general population, most of our participants were female students, which limits generalizability of the results. The online randomizer resulted in higher absolute means for each of the demographic and trait variables in the secure group and absolute standard deviation for trait paranoia, though there were no statistical differences in these variables between groups. Multiple testing raises the risk of type I errors; we did not use corrected p values given the novelty of the research, but this should be addressed in subsequent studies that are fully powered with larger samples. Our sample size was not large enough to test the impact of repeated priming on trait variables or examine possible moderation effects of trait variables, chiefly dispositional attachment anxiety and avoidance. The impact on trait measures would be of interest given the exploratory findings which suggest that repeated security priming may attenuate trait attachment anxiety but not avoidance (as found by Carnelley & Rowe [11]). Examining the moderation effects of trait attachment dimensions would clarify whether security priming works equally well for those high and low in attachment anxiety and in attachment avoidance.

## 5. Conclusions

The sustainability of effects on paranoia, mood and help seeking suggests that security priming can now be developed into an intervention for clinical populations. It will also be important to determine (a) the clinical (as opposed to statistical) impact [35] and (b) if the prime is most effectively implemented as a means of buffering (reducing reactivity to) or recovering from (facilitating a return to baseline following) the impact of stressors [69]—these would be valuable next steps and indicate how the intervention might be best utilised for people with clinical levels of paranoia.

## Figures and Tables

**Figure 1 brainsci-11-01257-f001:**
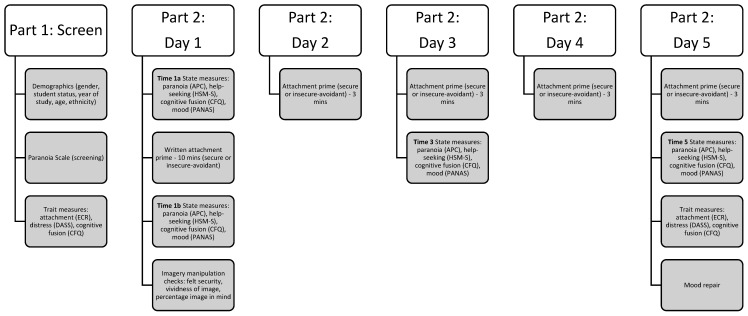
Study Procedure. Note: ECR = Experiences in Close Relationships Inventory; DASS = Depression Anxiety Stress Scale; CFQ = Cognitive Fusion Questionnaire; APC = Adapted Paranoia Checklist; HSM-S = State Help-Seeking Measure; PANAS = Positive and Negative Affect Schedule.

**Figure 2 brainsci-11-01257-f002:**
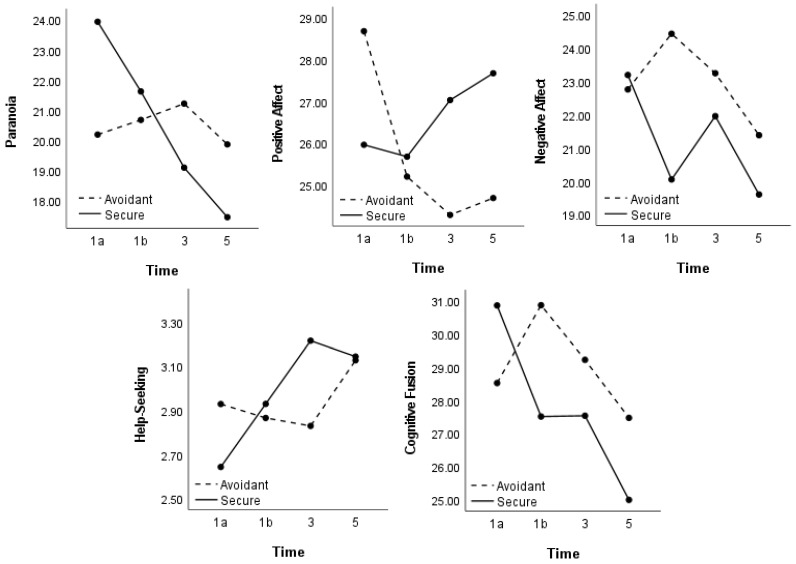
Change in state paranoia, affect, help seeking and cognitive fusion pre- and post-prime.

**Figure 3 brainsci-11-01257-f003:**
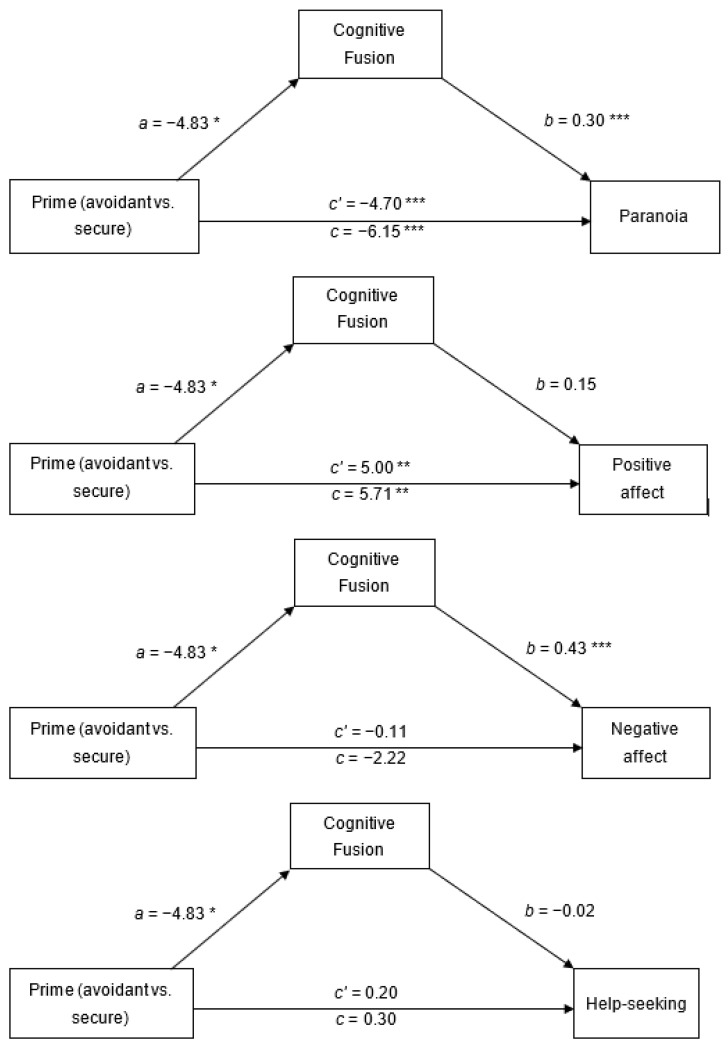
Mediation effects of secure vs. avoidant primes on paranoia, affect and help seeking via cognitive fusion. Note: *c*’ = direct effect; c = total effect; estimated path coefficients are unstandardized. * *p* < 0.05. ** *p* < 0.01. *** *p* < 0.001.

**Table 1 brainsci-11-01257-t001:** Descriptive statistics for trait characteristics.

	Secure Prime Group (*n* = 42)		Avoidant Prime Group (*n* = 37)	
	M	SD	M	SD
Trait paranoia	63.40	12.05	60.35	6.40
Trait depression	31.64	10.99	31.38	11.32
Trait anxiety	30.88	9.06	28.86	9.22
Trait stress	35.79	10.95	34.41	9.16
Trait cognitive fusion	36.24	8.49	33.11	8.00
Trait attachment anxiety	4.69	0.85	4.40	1.02
Trait attachment avoidance	3.92	1.00	3.79	0.93

**Table 2 brainsci-11-01257-t002:** Descriptive statistics for state measures pre- and post-prime.

	Secure Prime Group				Insecure-Avoidant Prime Group			
	Time 1a M (SD)	Time 1b M (SD)	Time 3 M (SD)	Time 5 M (SD)	Time 1a M (SD)	Time 1b M (SD)	Time 3 M (SD)	Time 5 M (SD)
Paranoia	23.93 (9.01)	21.62 (10.23)	19.10 (8.91)	17.45 (9.92)	20.19 (8.13)	20.68 (9.00)	21.22 (9.47)	19.86 (9.12)
Positive affect	25.95 (9.15)	25.67 (10.75)	27.02 (9.03)	27.67 (11.21)	28.68 (9.06)	25.19 (9.99)	24.27 (9.10)	24.68 (9.18)
Negative affect	23.19 (8.83)	20.05 (9.57)	21.95 (9.52)	19.60 (8.67)	22.76 (8.84)	24.43 (9.82)	23.24 (9.12)	21.38 (8.70)
Help seeking	2.64 (1.18)	2.93 (1.32)	3.21 (1.20)	3.14 (1.38)	2.93 (1.11)	2.86 (1.18)	2.83 (1.11)	3.13 (1.25)
Cognitive fusion	30.86 (9.12)	27.50 (10.98)	27.52 (10.49)	24.98 (10.51)	28.51 (11.10)	30.86 (9.95)	29.22 (10.09)	27.46 (10.63)

Descriptive statistics for state measures pre- and post-prime.

## Data Availability

Data for this research are available upon reasonable request.

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
