# Peer review of "The Impact of Repeated Attachment Priming on Paranoia, Mood and Help-Seeking Intentions in an Analogue Sample"

_brainsci, 2021, doi:10.3390/brainsci11101257_

Round 1
Reviewer 1 Report
Based on a sample of 79 adults scoring 53 or above on the Fenigstein & Vanable's paranoia scale, the authors examined the extent to which secure attachment priming can reduce state paranoia, negative affect, and cognitive fusion, while increasing positive affect and help-seeking. The sample size was chosen using a statistical power analysis. To answer the research question, the sample was randomly divided into a secure attachment priming group ("SAP," n=42) and an insecure-avoidant priming group ("ISAP," n=37). The resulting groups were treated accordingly over 4 consecutive days.
The “treatment” consisted of asking the test persons to visualize and then write about either a secure (SAP group) or insecure-avoidant (ISAP group) attachment relationship for 10 minutes. On each of the following four days the test persons received an email prompting them to rehearse the visualization and complete the writing task for 3 minutes. Thereafter, the state measures were repeated.
Using mixed-models of variance, the authors found that “security priming decreases paranoia, negative affect, and cognitive fusion, and increases positive affect and help-seeking, compared to insecure-avoidant priming”. And the authors concluded “given the sustainability of effects, security priming can be developed into a viable intervention for clinical populations”.
Comments
The authors assume that “secure attachment priming is effective in reducing paranoia and distress, and that cognitive fusion is likely to be a key causal mediating factor”, without providing sufficient evidence to support this assumption.
In their manuscript the authors understand under “paranoia” a continuum of unfounded interpersonal threat beliefs and distinguish between non-clinical, high-risk and clinical cases, without specifying, however, how they define (operationalize) these groups and where they draw the line between the groups. Therefore, the reader does not know how to interpret a PS score of 53 -- what is the difference in severity to the "high risk" group, or the clinical paranoia group? All this becomes particularly confusing when the authors later say (Discussion) “security priming reduces paranoia and negative affect in people with elevated and clinical levels of paranoia”.
Some 85% of the sample were 20-year-old female students, so that a methodological problem arises when a few 50-year-olds are also included in the sample. Also, it is probably to be worried that a larger part of the female participants are psychology students. The generalizability of the results is therefore very limited.
Table 1 casts doubt on the suitability of the authors’ online randomizer since the mean values in the left column are without exception larger than those in the right column (this looks highly-superrandom). Also striking is the large between-group difference regarding the standard deviation of "trait paranoia".
Habituation effects were not taken into account, which certainly play a role if the test subjects are always asked to do exactly the same 3-minute thing on 4 consecutive days. Therefore, to speak of "sustainability of effects" is not really convincing.
Another problem is the large number of statistical tests ("multiple testing"), which in many cases is the cause of non-reproducible results.
Moreover, “statistical significance” does not necessarily mean “practical relevance” which is of crucial importance in view of the intended clinical application. Regrettably, it is not clear from the manuscript how much of the observed variance is actually explained by the therapeutic intervention.
I think there is a lot to improve in this manuscript in order to achieve an acceptable scientific value.
Author Response
Please see the attachments (response to reviewers and track changed manuscript). Thanks.

Reviewer 2 Report
- This sentence in the Introduction requires a citation: "Even an individual who is generally high in anxiety or avoidance is likely to have had attachment relationships characterised by security, and so develop a secure internal working model."
- The following statement in the Abstract should be softened because, though intriguing, the findings are still provisional given that this is the first study to demonstrate cognitive fusion is a strong predictor of negative affect among individuals with non-clinical paranoia: "Now that we understand (1) the impact of security priming on paranoia, affect and help-seeking behaviours, (2) causal mechanisms, and (3) sustainability of effects, security priming can be developed into a viable intervention for clinical populations."
- On p. 2 of the Introductions, this sentence requires citation(s): "In addition to a dispositional (or trait) attachment style, adults hold relationship specific patterns based on previous attachment relationships. Even an individual who is generally high in anxiety or avoidance is likely to have had attachment relationships characterised by security, and so develop a secure internal working model."
- Suggest softening the language in the following sentence because mediational analyses support impact of attachment on paranoia but does not allow a causal conclusion to be drawn: "Our study design allows us to draw conclusions about the causal processes involved in the impact of attachment for people with paranoia."
Author Response

(The authors gave the same response as above.)
